# 🧑‍🍳 A cookbook for hardware-friendly implicit learning on static data

**Maxence Ernoult**◆
maxence@rain.ai

**Rasmus Kjær Høier** *
hier@chalmers.se

**Jack Kendall**◆
jack@rain.ai

## Abstract

The following aims to be a pragmatic introduction to hardware-friendly learning of *implicit models*, which encompass a broad class of models from feedforward nets to physical systems, taking static data as inputs. Starting from first principles, we present a *minimal* hierarchy of independent concepts to circumvent some problems inherent to the hardware implementation of standard differentiation. This way, we avoid entangling essential ingredients with arbitrary design choices by naively listing existing algorithms and instead propose the draft of a "cookbook" to help the exploration of many possible combinations of these independent mechanisms.

## 1 Problem statement

**Learning at equilibrium.** Given an input $x$, we want to find a set of model parameters $\theta$ which minimizes a given objective $\mathcal{O}$ defined over the model (hidden and output) variables $s(x, \theta)$, such that the model variables abides by some constraint $\mathcal{C}$. Namely:

$$\mathcal{P}_1: \quad \min_\theta J(x, \theta) := \mathcal{O}(s(x, \theta), \theta) \quad \text{s.t.} \quad \mathcal{C}(x, s(x, \theta), \theta) = 0. \tag{1}$$

Note that $s$ may implicitly contain several layers, e.g. $s = (s^{1^\top}, s^{2^\top}, \cdots, s^{N^\top})^\top$. Classically, $\mathcal{O}$ is some *cost function* $\ell$ measuring the discrepancy between the model prediction and some ground-truth data and $\mathcal{C}$ the "physical laws" that governs the substrate sustaining the model prediction *at equilibrium* – note that this implicit formalism encompasses both feedforward models (65), deep equilibrium models (4), as well as resistive networks governed by Kirchoff's laws (42; 96; 85). The problem defined in Eq. (1) is traditionally solved via first-order optimization by estimating the gradient over a minibatch $\mathcal{B}$ (106): $\overline{g}(\theta) := \mathbb{E}_{x \sim \mathcal{D}}[d_\theta J(x, \theta)]$. Therefore, solving $\mathcal{P}_1$ boils down to *how to compute* these gradients. We herein call a "circuit" the abstract physical system that may realize a given equilibrium condition.

**The Lagrangian method.** One way to solve $\mathcal{P}_1$ is by writing the associated *Lagrangian* functional $\mathcal{L}_1(s, \lambda, \theta) := \mathcal{O}(s, \theta) + \lambda^\top \cdot \mathcal{C}(s, \theta)$ and solving for the *Karush-Kuhn-Tucker* (KKT) conditions (89; 13). Loosely speaking, the *primal* feasibility condition $\partial_\lambda \mathcal{L}_1 = 0$ and *dual* feasibility condition $\partial_s \mathcal{L}_1 = 0$ generalize the notions of "forward pass" and "backward pass" respectively, and can be viewed as two circuits determining $s$ and $\lambda$ whose equilibrium satisfy:

$$\begin{cases} \mathcal{C}(s, \theta) = 0, \\ \partial_s \mathcal{C}(s, \theta)^\top \cdot \lambda + \nabla_s \mathcal{O}(s, \theta) = 0, \end{cases} \tag{2}$$

with the resulting gradient estimate reading as $g(\theta) = \nabla_\theta \mathcal{O}(s, \theta) + \partial_\theta \mathcal{C}(s, \theta)^\top \cdot \lambda$. We briefly mention that this can also be derived using the *Implicit Function Theorem* (IFT) (106). Here as well,

---

*Chalmers University of Technology, Sweden
◆Rain AI

Second Workshop on Machine Learning with New Compute Paradigms at NeurIPS 2024(MLNCP 2024).

note that *feedforward nets trained by backprop are a special case of Eq. (2)*. Yet, it appears from Eq. (2) that naively solving for $s$ and $\lambda$ comes with some challenges from the (high-level) hardware viewpoint: i) **the transport problem**: $s$ and $\theta$ need to be transported from the inference circuit to the error circuit; ii) **the memory problem**: $s$ needs to be stored; iii) **analytical derivatives**: $\partial_s \mathcal{C}$ potentially contains analytical derivatives which may have to be computed with high precision; iv) **forward locking**: in a hierarchical model, the circuit computing $s^k$ becomes idle when computing $s^{k+1}$ (Eq. (3)); v) **backward locking**: symmetrically, $\lambda_k$ cannot be solved before $\lambda_{k+1}$ (Eq. (6)); vi) **forward–backward synchrony**: $\lambda$ is computed *after* $s$.

## 2 Algorithm design & general methods

### 2.1 Crafting the constraints

**Hierarchical constraints.** As the variable $s$ may subsume a hierarchy of layers, one may want to split the original optimization problem into a hierarchy of $K$ problems, given $s := (s^{1\top}, \cdots, s^{K\top})^\top$ (16; 51; 31; 105; 75):

$$\mathcal{P}_2: \quad \min_\theta J(x,\theta) := \mathcal{O}(s,\theta) \quad \text{s.t.} \quad \mathcal{C}_1(s^1, s^0 := x, \theta^1) = 0, \cdots, \mathcal{C}_K(s^K, s^{K-1}, \theta^K) = 0 \quad (3)$$

Here, the model is split into a hierarchy of $K$ subcircuits (or "blocks") with parameters $\theta^k$, state $s^k$, which may comprise one *or multiple layers* ($K \leq L$), subject to the influence of the previous circuit through $s^{k-1}$. Note that for a given set of constraints, $\mathcal{P}_1$ *is generally not equivalent to* $\mathcal{P}_2$ – see § 3.6 for an example. In this case, solving for the KKT conditions of $\mathcal{P}_2$ yields:

$$\mathcal{C}_k(s^k, s^{k-1}, \theta^k) = 0 \tag{4}$$

$$\partial_{s^K} \mathcal{C}_K(s^K, s^{K-1}, \theta^K)^\top \cdot \lambda^K + \nabla_{s^K} \mathcal{O}(s,\theta) = 0 \tag{5}$$

$$\partial_{s^k} \mathcal{C}_k(s^k, s^{k-1}, \theta^k)^\top \cdot \lambda^k + \nabla_{s^k} \mathcal{O}(s,\theta) + \partial_s^k \mathcal{C}_{k+1}(s^{k+1}, s^k, \theta^{k+1})^\top \cdot \lambda^{k+1} = 0. \tag{6}$$

Note that $\mathcal{O}(s,\theta) = \ell(s^K, y)$ corresponds to the classical "end-to-end" supervised learning setting in which case the second term of Eq. (6) vanishes except for the last block $s^K$.

**Relaxed constraints.** When $\mathcal{C} = \nabla_s \mathcal{K}$ (see Section 3), $\mathcal{P}_1$ can be relaxed through an optimal value reformulation with a *fixed* Lagrangian multiplier $\beta^{-1}$ as (79; 105; 37):

$$\mathcal{P}_3: \quad \min_{\theta,s} \mathcal{O}(s,\theta) \text{ s.t. } \mathcal{K}(s,\theta) \leq \min_{s'} \mathcal{K}(s',\theta), \quad \mathcal{L}_3(s,\theta,\beta) := \mathcal{O}(s,\theta) + \frac{1}{\beta}\left(\mathcal{K}(s,\theta) - \min_{s'} \mathcal{K}(s',\theta)\right)$$

$$\tag{7}$$

The resulting Lagrangian $\mathcal{L}_3$, sometimes called "surrogate", is intimately tied to energy-based learning (88) (Section 3) and is especially convenient to express quantities as *finite* differences in $\beta$ which would otherwise appear as *exact* derivatives in $\beta$ starting from $\mathcal{P}_1$ (which amounts to send $\beta \to 0$). Importantly, this relaxation could also well be applied *at each level of the hierarchy of* $\mathcal{P}_2$ (37).

### 2.2 Picking an algorithm to estimate $\lambda$'s: standard approaches

**Implicit differentiation.** In spite of the aforementioned problems inherent to Eq. (2), one may still directly solve for $\lambda$ in combination with other tricks (see section 2.3). For feedforward models, $\partial_s \mathcal{C}^\top$ may be *explicitly* inverted with $\lambda = -\partial_s \mathcal{C}^{-\top} \cdot \mathcal{O}$ reducing to *backprop*. In other cases, $\lambda$ may be computed by *implicit differentiation* (44; 10; 30), which comes in many different flavors depending on the constraints at hand (14). Note that these techniques can be slightly adjusted to accommodate some constraints. For instance, to avoid weight transport between the inference and error circuits, we can equip the error circuit with its own "feedback" parameters $\omega$ such that the error signal satisfies: $\partial_s C(s,\omega)^\top \cdot \lambda + \nabla_s \mathcal{O}(s,\omega) = 0$ (52; 76; 1; 48). While randomly sampled $\omega$ are sufficient for shallow architectures and simple tasks, some extra alignment mechanisms are needed to have $\omega$ roughly approximate $\theta$ *without explicitly transporting it* (71; 6; 102; 1; 47).

**Another factorization of the gradient.** Note that $g(\theta)$ could also well be factorized as $g(\theta) = \nabla_\theta \mathcal{O}(s,\theta) + d_\theta s^\top \cdot \nabla_s \mathcal{O}(s,\theta)$, where $d_\theta s$ satisfies the equilibrium condition:

$$\partial_s \mathcal{C}(s,\theta) \cdot d_\theta s + \partial_\theta \mathcal{C}(s,\theta) = 0. \tag{8}$$

In this case, the error signal is carried *forward* through $\partial_s \mathcal{C}(s, \theta)$, however its memory usage scales cubically with the number of neurons. While low-rank approximations of Eq. (8) have been proposed in the context of RNNs to mitigate its memory cost (101; 98; 11), an extreme way to trade memory for time complexity would be by perturbing every single weight by a small amount $\epsilon$ and measure the resulting loss change, yielding:

$$\mathcal{C}(x, s^{\pm\epsilon,e_i}, \theta \pm \epsilon e_i) = 0, \quad d_\theta \mathcal{O}(s, \theta) \approx \sum_{i=1}^d \frac{1}{2\epsilon} \left( \mathcal{O}(s^{\epsilon,e_i}, \theta + \epsilon e_i) - \mathcal{O}(s^{-\epsilon,e_i}, \theta - \epsilon e_i) \right) e_i, \quad (9)$$

where $\{e_i\}_{i=1 \cdots n}$ denote the canonical basis in the weight space. However, Eq. (9) is still highly inefficient as it takes $2d$ forward passes per iteration to compute a single gradient.

**Forward-only learning with zeroth order optimization.** One sweet spot between Eq. (8) and Eq. (9), which has recently regained some popularity, is *zeroth order optimization* (ZO) (53). ZO techniques estimate a projection of the gradient $g(\theta)$ along some direction $u$ in the weight space by performing multiple forward passes:

$$u^\top \cdot g(\theta) = d_\epsilon \left( J(\theta + \epsilon u) \right) |_{\epsilon=0} \approx \frac{1}{2\epsilon} \left( \mathcal{O}(s(\theta + \epsilon u), \theta + \epsilon u) - \mathcal{O}(s(\theta - \epsilon u), \theta - \epsilon u) \right), \quad (10)$$

An unbiased estimate of the gradient can be obtained by averaging multiple such derivatives: $g(\theta) = \mathbb{E}_{u \sim \mathcal{N}(0,\sigma^2)} \left[ d_\epsilon \left( J(\theta + \epsilon u) \right) |_{\epsilon=0} \cdot u \right]$ (94; 7). However, the variance of this gradient estimate scales cubically with the number of model parameters (80), thereby restricting the applicability of ZO to the realm of models which are small enough (93; 35; 7; 26) or behaving as such, i.e. *pre-trained models* (56). One way to mitigate this problem is to perturb neurons instead of synapses (25; 66; 15; 43), namely computing projections of the error signal as: $v^\top \cdot \lambda = d_\epsilon \mathcal{O}(s^\epsilon, \theta)|_{\epsilon=0}$ with $s^\epsilon$ implicitly determined through $\mathcal{C}(s^\epsilon, \theta) + \epsilon v = 0$. Similarly, an unbiased gradient estimate is obtained as $g(\theta) = \nabla_\theta \mathcal{O} + \partial_\theta \mathcal{C}(s, \theta)^\top \cdot \mathbb{E}_{v \sim \mathcal{N}(0,\sigma^2)} \left[ d_\epsilon \mathcal{O}(s^\epsilon, \theta)|_{\epsilon=0} \cdot v \right]$. Going beyond, one can teach small auxiliary networks synthetizing such directions to output "good" weight directions $u$ instead of randomly sampling them (26), reducing the variance of the gradient estimate at the cost of increasing the bias (81; 93).

## 2.3 Other tricks which independently apply

**Greedy learning & loss design.** It appears from Eq. (6) that in general, an error signal $\lambda^{k+1}$ must be passed backwards to "unlock" the computation of $\lambda^k$. In order to parallelize learning across blocks entirely, a heuristic consists in shutting off the top-down error signal ($\lambda^{k+1} = 0$) and recreate an error signal through a *locally-defined* (supervised (8; 77; 9; 99; 29) or self-supervised (55; 103; 33; 92)) loss, which amounts to choosing $\mathcal{O} = \mathcal{O}_1(s^1, \theta^1) + \cdots + \mathcal{O}_K(s^K, \theta^K)$. In this case, the resulting block satisfies the exact same adjoint equation as that of the original learning problem (Eq. (2)) so that all gradient computation techniques herein presented may also apply *block-wise*. Alternatively, another solution to backward locking is to *estimate* $\lambda_{k+1}$ with auxiliary modules (41).

**Checkpointing & reversible models.** One way to mitigate the memory problem mentioned above (at the expense of compute) is to *simultaneously* compute $s$ and $\lambda$, which can be viewed as activation checkpointing (18). In models with an explicit layer hierarchy ($\mathcal{P}_2$), another special instantiation of activation checkpointing is possibly when using *reversible models* (21; 28; 17; 57): a given constraint $\mathcal{C}_{k+1}(s^{k+1}, s^k) = 0$ can be explicitly inverted as $\mathcal{C}_{k+1}^{-1}(s^k, s^{k+1}) = 0$ such that $s^k$ can be recomputed *backward* from layer $s^{k+1}$ instead of being stored, or recomputed *forward* from $s^0 = x$. This can be achieved for instance by splitting each block state $s^k$ into two, $s^k = (s_a^{k\top}, s_b^{k\top})^\top$, and defining $\mathcal{C}_{k+1} = (\mathcal{C}_{k+1}^a, \mathcal{C}_{k+1}^b)$, with dedicated transformations $f_a, f_b$, as (28):

$$\mathcal{C}_{k+1}^a(s^{k+1}, s^k) = s_a^{k+1} - s_a^k - f_a(s_b^k, \theta_a^k), \quad \mathcal{C}_b^{k+1}(s^{k+1}, s^k) = s_b^{k+1} - s_b^k - f_b(s_a^k, \theta_b^k) \quad (11)$$

**Pipelining.** When dealing with a hierarchical model ($\mathcal{P}_2$), the block $\mathcal{C}_k$ may become *idle* or "forward-locked" after passing $s^k$ to $\mathcal{C}_{k+1}$ until next input comes in. A solution to this is to push *multiple inputs in sequence* through the blocks, allowing them to process different inputs *in parallel*, e.g. $\mathcal{C}_k$ processes input $x_{p+1}$ while $\mathcal{C}_{k+1}$ processes input $x_p$, the same strategy applying *backwards* for the computation of the $\lambda_k$'s for different inputs (40). As naive pipelining may still maintain

idleness "bubbles", more elaborate schemes have been proposed (24), for instance by allowing each block to alternate between forward pass and backward passes for different inputs (73), yet at the cost of *gradient staleness* – $\lambda$ is computed with a different $\theta$ that the one used to compute $s$. This problem can be mitigated for instances by by maintaining different weight versions at each circuit (74).

## 3 Forward-only learning beyond zeroth order

### 3.1 Energy-based (EB) models & Energy-based Learning (EBL)

When the constraint of the optimization problem Eq. (1) derives from an energy function $\mathcal{K}$, energy-based learning (EBL) refers to a family of gradient computation algorithms which implicitly estimate $s$ and $\lambda$ using a *single* circuit (39; 34; 72; 5; 96; 86; 88). Namely, if there exists some scalar function $\mathcal{K}$ such that $\mathcal{C} := \nabla_s \mathcal{K}$, defining $\mathcal{F} := \mathcal{K} + \beta \mathcal{O}$ where $\beta \geq 0$ is some scalar value, then one can estimate $g(\theta)$ by having the *same* circuit relax twice to equilibrium with two different values of $\beta$ and subsequent "nudged states" $s_{\pm\beta}$(86; 46):

$$s_{\pm\beta} : \nabla_s \mathcal{F}(\pm\beta, s_{\pm\beta}, \theta) = 0, \quad g(\theta) = \frac{1}{2\beta} \left( \nabla_\theta \mathcal{F}(\beta, s_\beta, \theta) - \nabla_\theta \mathcal{F}(-\beta, s_{-\beta}, \theta) \right) + \mathcal{O}(\beta^2) \quad (12)$$

Eq. (12) is in stark contrast with Eq. (2), though being mathematically equivalent through $\lambda = d_\beta s_\beta|_{\beta=0}$ (86; 75): instead of estimating $s$ and $\lambda$ on two circuits, $s_\beta$ and $s_{-\beta}$ are estimated on a *single* circuit through "energy minimization", which is why $\mathcal{C}$ is said to be *energy based* (EB). The core intuition behind the magic of EB learning is that while error signals are usually carried "backward" through $\partial_s \mathcal{C}^\top$ (Eq. (2)), since we have $\partial_s \mathcal{C}^\top = \nabla_s^2 \mathcal{K} = \partial_s \mathcal{C}$, error signals can in this case be equivalently carried "forward" through $\partial_s \mathcal{C}$ through a small perturbation of $s$ along $\nabla_s \mathcal{O}$ of sufficiently small $\beta$. Finally note that Eq. (12) is only one many variants (88) to estimate $d_\beta (\nabla_\theta \mathcal{F})|_{\beta=0}$ (86), where there is trade-off between the number of $s_\beta$ being evaluated and the resulting bias (107; 45).

### 3.2 The importance of nudging

**The weak & strong nudging limits.** The most current EBL setting is when $\mathcal{C}$ describes the implicit model itself and $\mathcal{O}$ the cost function: $(\mathcal{C}, \mathcal{O}) = (F = \nabla_s E, \ell)$ where we have denoted $K = E$. In this case, the condition on the nudged state reads (86):

$$\nabla_s E(s_\beta, \theta) + \beta \nabla_s \ell(s_\beta, \theta) = 0 \qquad (13)$$

We call the *nudging* factor the scalar controlling the strength of the cost function $\ell$ in the definition of $s_\beta$. For this choice of $(\mathcal{C}, \mathcal{O})$, $\beta$ is the nudging factor and since theoretical guarantees of Eq. (12) hold for $\beta \to 0$, it corresponds to small $\nabla_s \ell$ perturbations, which is therefore called the *weak nudging limit*. Conversely, when swapping the objective and the constraint (64; 65), i.e. $(\mathcal{C}, \mathcal{O}) = (\nabla_s \ell, E)$, which amounts to swap $E$ and $\ell$ inside Eq. (13), one can see that $\beta^{-1}$ now gates the error signal $\nabla_s \ell$. Therefore in this situation, $\beta \to 0$ corresponds to *large* $\nabla_s \ell$ perturbations, which is therefore called the *strong nudging limit*. While it seems counter-intuitive that strong nudging solutions are relevant – the main goal of of Eq. (1) should be to minimize the loss subject to constraints on the network energy and not the other way around – they are global minimizers of $\theta \to \ell(\theta, s(\theta))$ for certain choices of $E$ (65). Strong nudging can enable learning in situations where the physical system at use is noisy by improving the signal to noise ratio for the teaching signal (64; 45).

**Nudging through adiabatic oscillations.** EBL classically operates in the weak nudging regime and estimates the error signal $\lambda = d_\beta s_\beta|_{\beta=0}$ through a discrete, two-steps finite difference procedure, as highlighted in Eq. (12). Another way to view the error signal is as the contour integration in the complex plane $\lambda = \frac{1}{T|\beta|} \int_0^T e^{i2\pi t/T} s_{\beta(t)}$ with $\beta(t) := |\beta| e^{i2\pi/T}$. Then, provided $T$ is sufficiently large compared to the characteristic time constant which governs the relaxation of $\mathcal{C}$ to equilibrium, the error signal can be computed through slow adiabatic oscillations of the system (5; 45; 3).

**Nudging heuristics.** The error signal $\beta \nabla_s \ell$ may be transmitted *instantaneously* to the rest of the network, or with a delay. Denoting $h(\theta, s)$ the model logits, some output controller dynamics $u$ may integrate the error signal inside the output layer as $\tau_u \dot{u} + u = -\beta \nabla_o \ell(h(s, \theta))$ and feed it back to the rest of the circuit (65). There also exists other types of nudging among the broader family of

*Contrastive Learning* (CL) algorithms. Denoting $y$ some label, the "nudged" hidden state $h_y$ can be defined by hard-clamping $y$ to the output units, $\nabla_s E(h_y, o = y, \theta) = 0$, while prescribing a weight update of the same contrastive form as Eq. 12 (72). An hybrid between weak nudging and target clamping consists of clamping the output to a *weakly nudged target*, resulting in a nudged hidden state $h^\beta$ defined as: $\nabla_s E(h^\beta, o = (1 - \beta)o_\star + \beta y) = 0$ where $o_\star$ corresponds to the free state ($\beta = 0$) of the output (96).

**Nudging synapses.** One may wonder whether neurons and parameters could play a symmetrical role, by having them *both* evolve throughout the gradient computation phase or satisfy an equilibrium condition. One common *post hoc* solution is to play on characteristic time constants by having parameters slowly integrate the *instantaneous* contrastive updates prescribed by Eq. (12) (22; 97; 45; 20), which however requires the detailed knowledge of the energy function at use. Another solution circumventing this constraint is to let *both* s and $\theta$ equilibrate in the same energy landscape, with control variables $u$ acting upon $\theta$ with strength $\alpha$ (87). When computing $s_{-\beta}$ (for instance), $u$ is adjusted to keep $\theta$ at its current value $\theta_t$ such that $u_t = \theta_t + \alpha \nabla_s \mathcal{F}(-\beta, s_{-\beta}, \theta_t)$. Therefore, keeping $u_t$ constant when computing $s_\beta$, $\theta$ equilibrates at: $\theta_{t+1} = u_t - \alpha \nabla_s \mathcal{F}(\beta, s_\beta, \theta_{t+1}) \approx \theta_t - \alpha \beta g(\theta)$.

### 3.3 Two phases or twice as many neurons?

**Lagrangian reparametrization.** One may usually regard the contrastive update Eq. (12) as a procedure performed on the *same* physical system whose state is measured at two different times with two different nudging values (e.g. $-\beta$, $\beta$), or a continuum of such. Yet, another view of Eq. (12) is to assume that the two nudged states are computed *simultaneoulsy by two different circuits* sharing the same parameters. An advantage of this implementation, which has been experimentally demonstrated with transistor-based synapses (20), is that it requires a single phase only instead of two, yet at the expense of area and complex engineering to enable parameter sharing. A different way to approach the same question is through the concept of *dyadic neurons* (36; 37): two variables $s_+$ and $s_-$ are such that their (convex) sum encodes the model prediction and their difference the error signal if they are critical points of the following reparametrized Lagrangian (37):

$$\widetilde{\mathcal{L}}_1(s_+, s_-, \theta) := \mathcal{L}_1\left(\alpha s_+ + (1 - \alpha)s_-, (s_+ - s_-)/\beta\right), \tag{14}$$

where $\mathcal{L}$ defined inside Eq. (2) and $\alpha \in [0, 1]$. While the resulting gradient computation algorithm is generally a reparametrization of implicit differentiation by construction, when the constraint at use inside Eq. (1) is energy based and $\beta$ is sufficiently small, $s_\pm$ can be simply construed as $s_{\pm\beta}$ of Eq. (12). This can be shown by performing this reparametrization on the *relaxed* Lagrangian $\mathcal{L}_3$ (Eq. (7)) with *fixed* $\beta$ (37), or equivalently approximated from Eq. (14) in the limit $\beta \to 0$.

### 3.4 Applying EBL to implicit models

**Weak nudging.** One of the ways to cast an implicit model $F$, which does not explicitly derive from an energy function, into an EB model is simply to employ the energy function $\mathcal{K} = E = \frac{1}{2}\|F\|^2$ (65). An important case, as one means to train feedforward nets by EP, is when $F(s, \theta) = s - f(\theta, s)$ where $\theta$ is typically lower block diagonal (27; 100; 67; 69). Eq. (12) for some $\beta$, in the weak nudging regime (($\mathcal{O}, \mathcal{K}) = (\ell, E)$), yields in this case:

$$\begin{cases} s_\beta = f(s_\beta, \theta) + \partial_s f(s_\beta, \theta)^\top \cdot (s_\beta - f(s_\beta, \theta)) - \beta \nabla_s \ell(s_\beta, \theta), \\ g(\theta) = \frac{1}{\beta} \partial_\theta f(s_\beta, \theta)^\top \cdot (s_\beta - f(s_\beta, \theta)) + \mathcal{O}(\beta) \end{cases} \tag{15}$$

A nice feature of Eq. (15) is that by construction, $s_\beta = f(s_\beta, \theta)$ when $\beta = 0$ such that the gradient can be estimated up to $\mathcal{O}(\beta)$ with a *single* nudged state. However, the presence of $\partial_s f(s_\beta, \theta)^\top$ signals the potential use analytical derivatives. To obviate this, one may instead estimate the required derivatives through finite differences of *feedback operators* (50; 82; 62; 63; 23; 61) as: $s_\beta \approx f(s_\beta, \theta) + g(s_\beta, \omega) - g(f(s_\beta), \omega)) - \beta \nabla_s \ell(s_\beta, \theta)$ where the feedback parameters may have to be learned so that $\partial_s g \approx \partial_s f^\top$ (23). However a problem remaining is that both $s_\beta$ and their "feedforward" prediction $f(s_\beta)$ need to be simultaneously maintained to implement Eq. (15). A solution is to use auxiliary variables $\psi$ which track the feedforward activity such that $\psi_\beta \approx f_\psi(s_\beta, \theta)$ and $s_\beta \approx f(s_\beta, \theta) + g(s_\beta, \omega) - g(\psi_\beta, \omega) - \beta \nabla_s \ell(s_\beta, \theta)$ (82).

**Recovering a least control problem in the strong nudging regime.** When in the strong nudging regime, the optimization problem Eq. (1) using $E = \frac{1}{2}\|F\|^2 = \mathcal{O}$ along with $\ell = \mathcal{K}$ acquires a very intriguing meaning, though the loss is not picked as the objective. Defining $\psi_\beta := -F(s_\beta, \theta)$, the learning problem amounts to find the set of parameters which minimizes the amount of *control* $\mathcal{O} = \frac{1}{2}\|\psi_\beta\|^2$ such that the controlled equilibrium $\psi_\beta + F(s_\beta, \theta) = 0$ minimizes the loss $\ell$, which works in practice on feedforward and implicit models more broadly (65). However, this interpretation is tied to the choice $E = \frac{1}{2}\|F\|^2$ and further investigation is needed to extend the applicability of strong nudging and its connection to control theory beyond this choice of energy function.

### 3.5 Handling nonlinearity inside EB models

When not handled cautiously, the application of Eq. 12 to implicit models yields activation derivatives (Eq. (15)). While neuroscientists, seeking for biologically plausible learning theories, gave birth to the aforementioned approximations of Eq. (15) and many others, these remain impractical for hardware-friendly learning. One *ad hoc* solution used in practice, when the inverse of the activation function is continuously invertible, is to separate the linear and nonlinear contribution inside the energy function as $E = G + U$ where $G$ is defined such that $\nabla_s G = \sigma^{-1}$ (39; 105; 36; 88). A more principled solution to handle $\sigma$ is to instead treat it as a *constraint* $\mathcal{C}_\sigma$ on the set of feasible configurations over which energy minimization is performed, which descent steps are *hard* projected onto(85). A "relaxed" version of this approach, somewhat bridging with the former solution, is through the lens of *mirror descent* using the Bregman divergence associated with $G$ (2): $s_{t+1} = \arg\min\{s^\top \cdot \nabla_s E(s) + \frac{1}{\eta}\Delta_G(s, s_t)\}$ with $\Delta_G(x, y) := G(x) - G(y) + \nabla G(y)^\top \cdot (x - y)$. Note that many other fixed point algorithms could be use for the same purpose (14).

### 3.6 Casting feedforward nets into hierarchical energy-based models

**An important example.** Another way to cast feedforward nets training by EBL algorithms is to break them into multiple, hierarchical energy-based constraints based on Eq. (3) (105). The following example, which shows how to do so, is also a good illustration of why the problems $\mathcal{P}_1$ (Eq. (1)) and $\mathcal{P}_2$ (Eq. (3)) are generally not equivalent. Taking $E := \sum_{k=1}^K E_k$ with $E_k(s^k, s^{k-1}, \theta^k) := G(s^k) - s^{k^\top} \cdot \theta^k \cdot s^{k-1}$, we consider $\mathcal{P}_1$ with $\mathcal{C} = \nabla_s E$ and $\mathcal{P}_2$ with $\mathcal{C}_k = \nabla_{s^k} E^k$. In this case, $s^k$ *is a single layer* and not a block of several layers. Setting the constraint to zero in these two cases yields:

$$\mathcal{P}_1 : \ s^k = \sigma\left(\theta^k \cdot s^{k-1} + \theta^{k+1^\top} \cdot s^{k+1}\right), \quad \mathcal{P}_2 : \ s^k = \sigma\left(\theta^k \cdot s^{k-1}\right) \tag{16}$$

In one case, the model retained at inference time is a *Hopfield model* and in the other one a *feedforward* model. This difference is instrumental to understand why works revolving around "contrastive learning" may actually apply to feedforward models through the $\mathcal{P}_2$ problem formulation (36). Likewise, the nudged state defined by Eq. (12) reads for any layer *until penultimate* the same as Eq. (16) for problem $\mathcal{P}_1$ and as follows for $\mathcal{P}_2$, taking $\beta$ sufficiently small (105; 75):

$$\mathcal{P}_2 : \ s_{\pm\beta}^k = \sigma\left(\theta^k \cdot s^{k-1} \pm \beta\theta^{k+1^\top} \cdot d_\beta s_\beta^{k+1}|_{\beta=0}\right) \approx \sigma\left(\theta^k \cdot s^{k-1} \pm \theta^{k+1^\top} \cdot \left(s_\beta^{k+1} - s_{-\beta}^{k+1}\right)/2\right), \tag{17}$$

Note that many other choices of energy functions are possible in order to learn feedforward nets as a hierarchical energy-based model (16; 51; 31; 105)

**Breaking the forward–backward synchrony.** As seen from Eq. (17), computing $s_{\pm\beta}^k$ still requires the storage of $s^{k-1}$. A trick is to notice that for sufficiently small $\beta$, $s^k \approx (s_\beta^k + s_{-\beta}^k)/2 := \bar{s}_\beta$ such that $s_\beta^k \approx \sigma\left(\theta^k \cdot \bar{s}_\beta \pm \theta^{k+1^\top} \cdot \Delta s_\beta^{k+1}\right)$ with $\Delta s_\beta^{k+1} := s_\beta^{k+1} - s_{-\beta}^{k+1}$. Implemented this way, the gradient corresponding to problem $\mathcal{P}^2$ with the above choice of energy function can be computed by *simultaneously* solving for $s_\beta^k$'s and $s_{-\beta}^k$'s (36). This implementation can also be derived by picking hierarchical constraints (Eq. (3)), relaxing these (Eq. (7)) and reparametrizing them (Eq. (14)) (37).

## 4 Looking ahead & take-aways

**Going out of equilibrium?** One of the ways to leverage equilibrium techniques towards learning from sequences is to treat the neuron *velocity* $\dot{s}$ as a variable of its own inside the energy function

(32; 90), or to put it differently to pick the *Lagrangian* of the physical system as an energy function such that the resulting equilibrium is an actual trajectory (84). Another exciting route beyond equilibrium models is to consider system which have physical access to their *adjoint* trajectory through *time-reversible*, Hamiltonian-based dynamics (54). This requirement can be regarded as the condition to be self-adjoint in the situation where the constraint $\mathcal{C}$ at use is a *trajectory* rather than an equilibrium state. In such a case, the associated algorithmic baseline is no longer implicit differentiation but the *continuous adjoint state method* (83).

**Forward-only learning beyond first order.** One may wonder if (and how) *second-order optimization* could be emulated using a *single* circuit. A second-order update generally prescribes $\Delta\theta_{2\text{nd}} \propto \mathbb{E}_{x\sim\mathcal{D}}[C(x,\theta)]^{-1} \cdot \mathbb{E}_{x\sim\mathcal{D}}[g(x,\theta)]$ where $\mathcal{C}$ is a curvature matrix, i.e. the loss *Hessian* or p.s.d. approximations thereof (58). As such, computing $\theta_{2\text{nd}}$ generally subsumes (*Hessian-free* approaches being a notable exception (60)): i) computing $g$, ii) computing $C$, iii) inverting $C$, iv) computing $\Delta\theta_{2\text{nd}}$. However, an interesting insight arises when $|\mathcal{D}| = 1$ (unit batch size), picking the Fisher information curvature matrix $C = (\partial_\theta s^\top \cdot \partial_\theta s)$ (59) and assuming $\mathcal{O}$ does not explicitly depend on $\theta$ for simplicity. Then, denoting $A^\dagger$ the *Moore-Penrose pseudo-inverse* of a matrix $A$:

$$\Delta\theta_{2\text{nd}} \propto \left(\partial_\theta s^\top \cdot \partial_\theta s\right)^{-1} \cdot \partial_\theta s^\top \cdot \nabla_s \mathcal{O} = \partial_\theta s^\dagger \cdot \nabla_s \mathcal{O}. \tag{18}$$

Therefore, moving to second order optimization here amounts to route the error signal $\nabla_s \mathcal{O}$ backward through the system's *inverse* rather than its adjoint. While this viewpoint has motivated several implementations of *self-invertible* (rather than *self-adjoint*, or equivalently energy-based) circuits (50; 62; 12; 63), it may not straightforwardly generalize to the mini-batch regime ($|\mathcal{D}| > 1$).

**Finding energy functions that map to physical systems.** Given a system described by the implicit mapping $\mathcal{C} = 0$, one can always pick $E = 1/2 \times \|\mathcal{C}\|^2$ (65), or even other variants picking other distances beyond the Euclidean metric, as a valid energy function. Also, the hierarchical energy-based setting depicted above offers even more flexibility to cast many feedforward architectures into an energy-based model. Beyond these tricks, other important computational primitives such as batch normalization or attention can also be construed as energy-based operations (38), yet to be investigated in the context of forward-only energy-based learning. So at first glance, "everything" seems to be energy-based and therefore trainable by the aforementioned forward-only techniques. However, all energy functions may not necessarily map to a physical system, namely a lot of these energy functions may *not* be physical. For instance, the fact that layered resistive networks can compute MLP operations at equilibrium *and* are governed by the minimization of an energy function that is itself the pseudo-power of these circuits (42; 85) is truly remarkable. Therefore, finding an energy function which realizes some core computational primitive *while being physically feasible* is difficult.

**Analog-digital codesign.** While *fully analog* learning experimental proof-of-concepts have been realized on small systems (20; 104), bringing analog systems at chip scale inherently requires digital circuitry for registers, tile-to-tile routing, analog non idealities mitigation (ADCs & DACs), and more fundamentally for the implementation of analytical and possibly non-weight stationnary operations (95; 49; 104). Beyond this practical aspect, one may wonder if we even *should* map all operations into analog. For instance, attention operations are known to be memory-bounded both at inference and training time when using large context windows rather than bottlenecked by the digital compute itself (91; 19). So it is unclear whether or not analog computing may necessarily bring an advantage for this particular example. Overall this observation calls for digital-analog codesign, and to a further extent for the design of new theoretical building blocks that would account for both digital *and* analog parts of a circuit with associated learning algorithms (75).

**Conclusion.** This "cookbook" is an attempt to complement recent algorithm unification papers (105; 106; 68; 78; 70) using pragmatic, ingredient-based logics. For instance, "Predictive coding" (100) amounts to apply Eq. (12) with a specific choice of energy function (3.4) and nudging scheme (3.2). Likewise, the "Forward-forward" algorithm (33) is a greedy learning strategy, with each block comprising a single layer, using local self-supervised loss (2.3). Conversely, one may want to split an architecture into blocks (Eq. (3)), apply standard implicit differentiation in some (2.2, §1), ZO in others (2.2, §1), along with a pipelining mechanism (2.3) depending on the constraints at hand, which does not correspond to any known algorithm in the literature.

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
