# OpenReview forum: "A cookbook for hardware-friendly implicit learning on static data"
_NeurIPS.cc/2024/Workshop/MLNCP — MLNCP Poster_

### Official Review · Reviewer_xd1L · 2024-10-04
**Strong review paper with a unified mathematical framework**

**Rating:** 8
**Confidence:** 3

**Review:**

# Summary
The presented paper is a review of hardware-friendly learning methods for implicit models. The strength of this paper is its unified presentation of these methods with a focus on solving challenges in hardware implementations such as the weight-transport problem and other related problems.
# Discussion
The topic of the paper is highly relevant for the workshop. Novel hardware accelerators often times come with additional constraints in contrast to general purpose computers like GPUs. The paper formalizes many such important constraints in a unified optimization framework and discusses this formulation in the context of a large body of relevant literature.

The presentation is both conceptually and mathematically clear and concise. The paper will be a valuable addition to the workshop. If alternative learning methods want to become more competitive with backprop at realistic problem sizes, a framework for casting the promises and limitations of these methods  might help overcoming some of the scaling issues that these methods still have. The paper might serve as a basis of discussion for future theoretical and empirical evidence.

---

### Decision · Program_Chairs · 2024-10-10

Accept (Poster)